# From growers to patients: Multi-stakeholder views on the use of, and access to medicinal cannabis in Australia

**Daniel Erku**[1,2]*, **Lisa-Marie Greenwood**[3], **Myfanwy Graham**[4,5], **Christine Mary Hallinan**[6], **Jessica G. Bartschi**[4,7,8], **Elianne Renaud**[4], **Paul Scuffham**[1,2]

**1** Centre for Applied Health Economics, School of Medicine, Griffith University, Brisbane, Australia, **2** Menzies Health Institute Queensland, Griffith University, Gold Coast, Australia, **3** Research School of Psychology, The Australian National University, Canberra, ACT, Australia, **4** Australian Centre for Cannabinoid Clinical and Research Excellence, University of Newcastle, Newcastle, Australia, **5** Centre for Drug Repurposing & Medicines Research, School of Medicine and Public Health, University of Newcastle, Newcastle, Australia, **6** Department of General Practice, Faculty Dentistry Medicine and Health Science, University of Melbourne, Melbourne, Australia, **7** School of Psychology, Faculty of the Arts, Social Sciences and Humanities, University of Wollongong, Wollongong, Australia, **8** Illawarra Health and Medical Research Institute, University of Wollongong, Wollongong, Australia

\* d.erku@griffith.edu.au

**Data Availability Statement:** All underlying data used in this study is publicly available at Australian government parliament website, and can be accessed at https://www.aph.gov.au/

## Abstract

### Background

Patient interest in the use of cannabis-based medicines (CBMs) has increased in Australia. While recent policy and legislative changes have enabled health practitioners to prescribe CBMs for their patients, many patients still struggle to access CBMs. This paper employed a thematic analysis to submissions made to a 2019 Australian government inquiry into current barriers of patient access to medical cannabis.

### Methods

We identified 121 submissions from patients or family members (n = 63), government bodies (n = 5), non-government organisations (i.e., professional health bodies, charities, consumer organisations or advocacy groups; n = 25), medical cannabis and pharmaceutical industry (n = 16), and individual health professionals, academics, or research centres (n = 12). Data were coded using NVivo 12 software and thematically analysed. The findings were presented narratively using a modified Levesque's patient-centred access to care framework which includes: i) appropriateness; ii) availability and geographic accessibility; iii) acceptability; and iv) affordability.

### Results

Submissions from government agencies and professional health bodies consistently supported maintaining the current regulatory frameworks and access pathways, whereas an overwhelming majority of patients, advocacy groups and the medical cannabis industry described the current regulatory and access models as 'not fit for purpose'. These differing views seem to arise from divergent persepctives on (i) what and how much evidence is

Parliamentary_Business/Committees/Senate/
Community_Affairs/Medicinalcannabis.

**Funding:** The author(s) received no specific
funding for this work.

**Competing interests:** The authors have declared
that no competing interests exist.

needed for policy and practice, and (ii) how patients should be given access to medical cannabis products amidst empirical uncertainty. Notwithstanding these differences, there were commonalities among some stakeholders regarding the various supply, regulatory, legislative, financial, and dispensing challenges that hindered timely access to CBMs.

## Conclusions

Progress in addressing the fundamental barriers that determine if and how a patient accesses and uses CBMs needs i) a 'system-level' reform that gives due consideration to the geographic disparity in access to prescribers and medical cannabis, and ii) reframing societal and health professional's views of CBMs by decoupling recreational vs medical cannabis.

## Introduction

Community interest in the use of cannabis for therapeutic purposes has increased within Australia and internationally [1, 2]. The rapid growth in national demand for access to cannabis for therapeutic use has largely been driven by patient or consumer advocacy groups, increased media attention, and a growing yet still incomplete body of research examining cannabis' treatment efficacy [1]. Legislative changes at the level of Australian state and federal government have occurred over the past decade to legalise the use of medical cannabis, and to regulate its prescribing and distribution. In 2014, the state of New South Wales legalised prescription access to medical cannabis on compassionate grounds to individuals with terminal illnesses [3], and in 2016 Victoria legalised medical cannabis with no constraints [4]. The most significant legislative changes followed, when in late 2016, the Australian Federal government amended the Narcotics Drug Act (1967) to enable the cultivation, production, and manufacture of medical cannabis [5, 6].

Alongside these legislative changes, Australia (like many other jurisdictions worldwide) has witnessed an increase in patient demand for access to medicinal cannabis and renewed scientific interest in its therapeutic use. A preliminary search of Australian New Zealand Clinical Trials Registry showed that, as of December 2021, over 50 observational and randomised controlled trials (RCTs) of medicinal cannabis products for various indications have been registered. While research into the use of medicinal cannabis products continues, Australian policy and legislative changes have enabled health practitioners to prescribe medicinal cannabis to their patients typically before products have satisfied the standards of quality, safety, and efficacy required for registration by the Therapeutic Goods Administration (TGA). Despite the availability of numerous unapproved medicinal cannabis products for prescription by health practitioners [7], currently only two medicinal cannabis products are listed on the Australia Register of Therapeutic Goods (ARTG)—Sativex® (nabiximols) and Epidyolex® (cannabidiol (CBD) 100 mg/mL) [8]. In the case of unapproved medical cannabis products, clinicians need to apply to TGA via the Special Access Scheme (SAS) Category A notification pathway, SAS Category B application pathway, or as an Authorised Prescriber (AP; requiring jurisdictional and/or TGA pathways). SAS Category A is a notification pathway that may be used by a prescribing registered health practitioner (including nurse practitioner) to access unapproved therapeutic products including medical cannabis via importation for patients defined as seriously ill. SAS-B is an access pathway through which a registered health practitioner applies to the TGA for approval to prescribe an unapproved medical cannabis product for a patient

under their care [1]. In the case of Authorised Prescribers, approval may be granted to access an unregistered product under the 'established history of use pathway' for up to 5 years, pending the endorsement of this timeframe, or the 'standard pathway' requiring approval from a human research ethics committee or endorsement by specialist college (accurate as of November 2021; [9]).

Reflective of the complexity of accessing unapproved products, in November 2019, the Australian Senate referred an inquiry to the Senate Community Affairs References Committee into current barriers to patient access to medical cannabis in Australia [10]. This review was established in response to anecdotal evidence that patients continued to report various difficulties in accessing medical cannabis since its legalisation [10]. During the inquiry, the committee called for public submissions to inform current barriers to accessing medical cannabis across the supply chain. During 69 days of consultation in 2020, multiple stakeholders, including patients, clinicians, and organisations, submitted their perspectives relating to the terms of reference. The recommendations from the Committee's report highlighted key priority areas in refining and expanding Australia's regulatory processes. A significant and ongoing challenge for the sector is how to draw forward a unified strategic approach to medical cannabis access from the various, sometimes divergent, stakeholder perspectives. Determining key similarities and differences in stakeholder views, priorities, and preferences, is needed to balance inclusive and equitable access to medical cannabis with safety and regulatory requirements. Despite the rich and descriptive accounts provided through the inquiry, how the views of different stakeholders, including the public health community in Australia, have framed the medical cannabis-related policy debate has not been systematically explored. Using the submissions from stakeholders, this study aimed to (1) identify the views of multiple stakeholders regarding medical cannabis; (2) identify the policy frameworks and preferred regulatory options recommended by stakeholders, and identify the associated discourses used to describe them; and (3) elucidate the ways in which scientific evidence about medical cannabis has been framed to justify the proposed regulatory frameworks and changes relative to medical cannabis access.

## Methods

### Data sources

In December 2019, the Senate Community Affairs References Committee invited the public and interested organisations to make submissions to an inquiry into the current barriers to patient access to medical cannabis in Australia. The terms of reference sought comment across major areas of Australian's medical cannabis landscape, including:

- The appropriateness of Australia's regulatory frameworks for medical cannabis products, inter-state differences in interpretation and access, and comparison to international best practice models;

- Regulatory barriers to accessing medical cannabis (including those in rural and remote areas), and its impact on the mental and physical wellbeing of patients;

- Financial barriers and the suitability of the Pharmaceutical Benefits Scheme (PBS) for subsidising patient access to medical cannabis; and

- Information sources for health professionals, availability of and need for education and training.

The Inquiry received 146 submissions from government and non-government organisations, professional health bodies, health charities, advocacy groups, medical cannabis industry and members of the public (S1 File). The Standing Committee also conducted a public hearing

in January 2020, the transcripts of which were later published. This study analysed both written submissions and, where submission authors participated in a public hearing, the relevant section of the transcript from the hearing.

## Data extraction and analytical framework

Data relating to barriers to and solutions for patients accessing medical cannabis, and arguments made for and against adopting specific regulatory approaches, were coded using NVivo version 12 software (QSR International Pty Ltd, Melbourne, Australia), then thematically analysed using the Framework Method [11]. The Framework Method, originally developed in the 1980s by the UK National Centre for Social Research, is a type of thematic analysis which involves developing a matrix of structured thematic categories into which the data is coded and analysed either by case (in this case individual submissions) and by code (i.e., a descriptive or conceptual label is assigned to excerpts of the submissions in the coding process). This method of thematic analysis was chosen for this study as it enables analytical themes to be developed both from the inquiry's terms of references and from the narratives of individual submissions.

The analysis included five steps. Firstly, a team of researchers (DE, LG, and MG), all from different research fields, read through a randomly selected subset of submissions and transcripts from the public hearing (five submissions each). These submissions were then used to identify common themes and come up with initial coding framework (step 2). Following this, the whole team met to discuss key emerging themes and develop, revise, and refine an initial coding framework (step 3). The initial analytical framework and pre-specified codes generated through independent coding, discussion and consensus were found to be similar to the patient-centred access to care framework previously developed by Levesque *et al*. [12]. Thus, data from all submissions was extracted (by one author–DE) and coded using Levesque's framework which includes the following dimensions or themes (step 4): (1) appropriateness; (2) availability and geographic accessibility; (3) acceptability; and (4) affordability. Appropriateness refers to the suitability of current regulatory frameworks and access pathways for medical cannabis and the extent to which patient needs for access were met by these frameworks. Data relating to the impact of patient location (according to state or territory and rural or remote areas) has on access to prescribing health professionals and/or medical cannabis products were coded under "availability and geographic accessibility". The "acceptability" code relates to the cultural, ethical, and social factors that determine health professionals' preparedness to prescribe medical cannabis and for patient use for therapeutic purposes. These factors are underpinned by complex social constructs that include patient stigma associated with cannabis use, attitudes of consumers and health professionals towards medical cannabis, and the current regulatory frameworks. The "affordability" refers to costs associated with medical cannabis and ability to pay according to income. Thematic analysis was then conducted on the coded data, guided primarily by our research objectives (i.e., explore views of multiple stakeholders regarding medical cannabis, and recommended regulatory and policy options to improve access) and by emergent themes generated inductively from the data [13] (step 5).

## Results

### Sample overview and stakeholder categories

A total of 121 submissions were included in the analysis: 63 from patients or family members, 5 from government bodies, 25 from non-government organisations (i.e., professional health bodies, charities, consumer organisations or advocacy groups), 16 from medical cannabis and

**Table 1. Stakeholders included in the analysis, n = 121.**

| Stakeholder group | N (%) |
|---|---|
| **Governmental organisations** | 5 |
| **Non-governmental organisations** | |
| Professional health bodies | 7 |
| Health charities, patient and consumer advocacy groups | 18 |
| Pharmaceutical and medicinal cannabis industry | 16 |
| **Member of the public** | |
| Health professionals, academics, or research centres | 12 |
| Patients and caregivers | 63 |

pharmaceutical industry, and 12 from individual health professionals, academics, or research centres (Table 1). Submissions were also supplemented with interview transcripts from the public hearing. We excluded confidential (not publicly available) submissions (n = 21) and supplementary attachments (n = 4). The website link to the original submissions publicly available on the Australian government parliament website are provided in S1 File.

## Use of and access to medical cannabis

Submissions from multiple stakeholders into the barriers for access to medical cannabis in Australia contained a range of arguments for and against various regulatory and access frameworks for medical cannabis. Submissions from professional health bodies, health charities and government agencies consistently supported maintaining the current regulatory frameworks and access pathways. Conversely, submissions from patients, patient advocacy groups, and the medical cannabis industry tended to describe the current regulatory and access models as 'not fit for purpose'. Given the nature of the inquiry, there were instances of submissions that predominately centered on advocacy for increased access and easing of costs and other regulatory burdens for patients. Some of these included proposed amendments to regulation and justification for these recommendations. However, analysis in this study focussed more on the thematic debates emerging within the discourse and their implications for the national framework. These negotiations seem to arise from divergent persepctives on (i) what and how much evidence is needed for policy and practice, and (ii) how patients should be given access to medical cannabis products amidst empirical uncertainty.

## The role of evidence in informing policy and clinical practice: Why, what, and how much?

**What counts as evidence?.** The analysis of the submissions and public hearing transcripts revealed a divergent viewpoint among stakeholders on the *level* and *credibility* of evidence that is needed to support changing and/or maintaining the current regulatory framework regarding medical cannabis. Submissions from government agencies and public health communities (professional health bodies and charities) discussed the importance of evidence-based policy making and asserted that their policy positions are based on the latest scientific evidence. They tended to suggest that there is a paucity of current clinical evidence in terms of the safety and efficacy of medical cannabis **(see *Submissions 2, 8, 119, 129)*,** and highlighted that further research is required to ensure future recommendations around the use of and access to medical cannabis is evidence based. This included the need for well-designed clinical trials with longer-term follow-up periods.

"*There are very few well designed clinical trials using medicinal cannabis resulting in limited evidence for its use in successfully treating different medical conditions, or on effective forms and dosages.*" (**Northern Territory Government**, **Submission 2; page 2**)

"*The development of a sound evidence base remains a critical enabler to ensure safe and effective use of medicinal cannabis in chronic pain and requires further research and investment.*" (**Pain Australia, Submission 129; page 6**)

Submissions from the health and medical field raised concern about approaches that introduced medical cannabis as a theraeputic option without considering the lack of clinical evidence on safety and efficacy, and other best practice treatment and management.

"*The gaps are substantial in current knowledge about the dose, delivery of different products, therapeutic use as add-on therapy or stand-alone therapy in the treatment of a broad spectrum of conditions and diseases. This poses an unacceptable risk in my view to changing the current requirements of registration that have a remarkable track record.*" (**Professor James Angus, Chair of the Australian Advisory Council on the Medicinal Use of Cannabis' Submission 53; page 2**)

Another finding was the inconsistent use of anecdotal evidence. Submissions from the health and medical professions which prioritised clinical evidence contrasted strongly against consumer and patient accounts. For example, the Australian Medical Association reported in their submission that some of their members were reluctant to prescribe medical cannabis because of the concern that it was introduced "based on anecdote and public opinion with the expectation that the science to catch up in due course" (**Australian Medical Association**, **Submission 24; page 5**). In contrast, patients, patient or consumer advocacy groups, and the medical cannabis industry described personal experiences of using medical cannabis within the therapeutic context (i.e., anecdotal evidence) to support a more permeable regulatory model. While the need to incorporate patients' preferences into medical decision making has been highlighted by various stakeholders as a key component of patient-centred care, this has not been translated to decisions around access to and use of medical cannabis. For example, some patients, families, and caregivers reported using medical cannabis due to the significant financial distress, poor symptom control or intolerable adverse effects from conventional therapies (e.g., opioid dependence). One patient described medical cannabis as "a gateway out of the hopelessness of opioid dependence. . . alcoholism and addiction to prescription benzos" (**Submission 132, page 1**). Yet, what patients described as their lived experiences was considered by many stakeholders as 'anecdotal' evidence, and it was argued that they should not be taken into consideration in the policy making process.

"*What patients are reporting 'anecdotally' is their lived experience. I think it more sensible in the complex space of medicinal cannabis to instead consider a 'risk vs benefits' approach and to make care related decisions accordingly because regardless of political and medical opinions, thousands of Australian patients are using cannabis medically and illegally.*" (**Individual; Submission 65; page 4**).

**Patient access amidst empirical uncertainty.** Whilst the need for high quality evidence on the safety and efficacy of medical cannabis products was widely acknowledged, there was a shared sense of recognition by stakeholders that there needs to be a way for medical cannabis to be managed and accessible to patients (even if conditionally) within the therapeutic context.

However, stakeholders were largely divided when it came to *how* patients should be given access to medical cannabis products, particularly given the growing yet still incomplete evidence. The view of some professional health bodies and government agencies was that medical cannabis products should be reserved as 'treatment of last resort' to a subset of patients with life-limiting illness.

> "*Despite the lack of a current evidence-base, medicinal cannabis may be considered an option of last resort for chronic pain management where a range of other therapies have been exhausted.*" **Pain Australia, Submission 129, page** 6

Again, clinical and governmental perspectives were not typically reflective of the views of patients and patient or consumer advocacy groups who regarded cannabis-based medicines as 'first resort rather than a last resort' (**e.g.,** *Submission 136, page 2*) and advocated for expedited access while Therapeutic Goods Administration registration is pending. This regulatory and access preference for patients using medical cannabis was reflected for conditions ranging from life threatening illness such as cancer to illnesses such as anxiety or short-term or chronic insomnia.

In addition, the ways in which stakeholders have defined and framed 'therapeutic use of cannabis' (also referred to in the submissions as 'cannabis-based medicines' or 'medicinal cannabis') seemed to have influenced their policy or practice recommendations. Submissions from some pharmaceutical industry entities with a commercial interest in the therapeutic use of particular products of cannabis (e.g., cannabis products registered in ARTG) have framed 'medicinal cannabis' as consumer products that lacked sufficient empirical evidence to be considered an approved medicine.

> "*Non-regulatory approved cannabis-based products ("medicinal cannabis") lack all or many of the important characteristics of an approved medicine, including stability, batch to batch consistency and label accuracy, and therefore the evidence base of these individual products is limited*" (**GW Pharmaceuticals, Submission 119; page 10**).

On the other hand, this submission framed 'cannabis-based medicines' as high quality, regulatory-approved 'pharmaceutical' grade products, and as such these products cannot be equated with or replaced by 'medicinal cannabis' products via product substitution. The reasoning provided for this was that 'not all products can be expected to have the same safety profile, therefore the safety data from one product cannot be extrapolated to another' (**GW Pharmaceuticals, Submission 119; page 10**). However, this notion was seen unfavourably by many patients and advocacy groups. Patients discussed that such framing stemmed from and largely shaped by the government's preconceived 'anti-cannabis' ideology, the financial conflict of interest between pharmaceutical and medicinal cannabis industry, and Australia's 'pharmaceutical lobbying culture' (**see Submission 91; page 9**).

> "*I believe it is mostly born out of ignorance, kickbacks from Big Pharma and personal belief systems which hinder an evidenced based approach to the application of Medicinal Cannabis in the area of medicine and in other allied health areas.*" (**Individual; submission 65; page 2**).

### Barriers to and solutions for addressing medical cannabis access

A significant proportion of patients noted that they had experienced obstacles in accessing medical cannabis products, and many stakeholders discussed various supply, regulatory, legislative, financial, and dispensing challenges that hindered timely access to medical cannabis

products. Using Levesque's modified patient-centred access to healthcare framework [12], we categorised these access issues and recommended solutions into i) appropriateness, ii) availability and geographic accessibility, iii) affordability, and iv) acceptability.

## "Not fit for purpose": Appropriateness of the current model in meeting patients' needs

Appropriateness of current regulatory frameworks and its responsiveness to the evolving medical cannabis market has been extensively discussed by stakeholders. While the current access schemes were considered by government agencies and professional health bodies as best-practice frameworks to ensure patient safety and informed prescribing decisions, many stakeholders including patients, patient advocacy groups and medical cannabis industries were concerned about the responsiveness of these frameworks to the evolving medical cannabis market, in terms of the diversity of patients and product preferences. The current access pathways were described by these latter groups as 'too convoluted, complex and barrier forming', and were thought to have put undue financial and psychological burden for people with low socio-economic status, older people reliant on pensions/disability allowances and those who live in rural and remote areas.

> "*No patient can find anything praiseworthy in a scheme that is deliberately crafted to present a daunting minefield of barriers impossible to negotiate, and, for the accepted few, culminating in an extortionately priced limited-range product so dilute and watered-down as to be next to useless.*" (**Individual; Submission 100; page 1**).

In addition to the reference to regulatory hurdles above, stakeholders regularly raised recurrent practical challenges associated with each of the access pathways. Submissions from patients, advocacy groups and the medical cannabis industry reiterated that one of the main frustrations associated with SAS-B scheme was the arduous and time-consuming process that health professionals have to go through for writing a prescription. These includes familiarising with the scheme, collating necessary evidence and clinical justification, and completing the actual application. When discussing these limitations of the SAS-B access scheme, submissions from multiple stakeholders noted the scheme's inflexibility to accommodate the unique features of medical cannabis products and need for reapplications (where a specific product formulation was not effective or not available) was attributed to further strain placed on patients by exposing them to unnecessary delays and financial stress **(e.g., see Submission 15, 22).**

It was indicated that the use of an AP scheme by prescribers had been limited mainly due to the challenges of getting approval from a human research ethics committee or endorsement from a specialist college **(e.g., see Submission 6; 25).** This is further complicated by the fact that none of the main professional Australian health bodies, including the Royal Australian College of General Practice or the Royal Australian and New Zealand College of Psychiatrists, endorse applications for authorised prescription of medical cannabis.

> "*Medical Practitioners have stated that they maintain greater clinical freedom in selecting the most appropriate medication for their patients from a wider range of products and cost points by choosing not to become Authorised Prescribers.*" (**Epilepsy Australia, Submission 22; page. 7**).

**Driving and medical cannabis: A legal quandary.**   Under the current Australian drug driving policy, driving while having detectable levels of tetrahydrocannabinol (THC;

dronabinol) in the body is considered an offence, and patients may be subject to prosecution, fines, and loss of licence [14, 15]. The current drug driving policy as it relates to legally pre-scribed cannabis was considered by many stakeholders as a significant barrier to legitimate patient access to medical cannabis, especially to patients in rural and remote locations.

> "*Patients who have a prescription may not want to fill it, because they have a job in which they have to drive every day and they would be at risk of prosecution for doing something which they took steps to do lawfully.*" **Dr Nicoletti,** *Committee Hansard***, page. 17.**

Stakeholders put forward various policy solutions, ranging from exemption from prosecu-tion for patients prescribed cannabis for therapeutic purposes to changing drug testing policy from a 'zero-tolerance' towards one that reflect the level of impairment.

**Illicit markets and self-cultivation.**   A significant proportion of patients who made sub-missions reported that they resorted to illicit markets (also referred to as 'black', 'grey' or 'green' market) to source their medical cannabis products, or reported self-producing medical cannabis for personal use, often without a license to produce medicinal cannabis for that pur-pose *(see submissions 4, 6, 13, 29, 44, 80, 91, 110).* The main reasons specified for resorting to illegal access routes included: the complexity of the legal access system; the 'unjust' regulatory restriction imposed on patients to utilise only medical cannabis products currently available on the market, which are perceived to be 'ineffective' and/or expensive; and the onerous pro-cess required to switch between products.

> "*There is no doubt Australia has restricted cannabis and cannabis product access to the point where the current model actually drives patients to the alternative market and self-supply. If the current model was in place to protect patients self-evidently it has been an abject failure.*" *(Individual; Submission 80; page 5*).

> "*Black market Cannabis costs about $250 an ounce in Australia. Legal Cannabis for medical use in Australia costs $1300–1500 for the 1st month and $800 a month or more after that. With no subsidy it is hard for many to justify the legal route.*" *(Individual; Submission 91; page 11*).

However, many patients who opted to source their cannabis from illicit markets felt that they had no other choice and yet, at the same time expressed concerns about the legal reper-cussions and safety issues associated with sourcing medical cannabis products from illicit mar-kets (*see submissions 4, 22, 36).* The safety concern was also voiced by public health community, such as the Australasian College of Nutritional and Environmental Medicine, that such unregulated products could "contain contaminants such as heavy metals and pesticides, and that they may not contain the amount of active constituents they purport to." *(Austral-asian College of Nutritional and Environmental Medicine, Submission 29, page. 19).* While a few submissions recommended amnesty be introduced for current users of illicit cannabis products for medical reasons *(Medical Cannabis Users Association, Submission 9, p. 21; Sub-mission 36, p. 10)*, others argued for broader decriminalisation of personal cannabis cultiva-tion and use for medical purposes *(Submission 6, pages 10–11; Submission 21, page 6; Submission 76, page 5; Submission 98,).*

## Availability and geographic accessibility

**Urban–rural and inter-state disparity in access to prescribers and medical cannabis.** Submissions from multiple stakeholders particularly patients, advocacy groups and patient

representatives, discussed the challenges of accessing a health professional who is both willing and able (in terms of knowledge, time, and legal permission) to prescribe medical cannabis products in rural, regional, and remote locations *(e.g., see submissions 15, 25, 35, 48, 116).* Highlighting equity considerations, submissions often reiterated that the opportunity for a low-income person living in rural area to access a medical cannabis product cannot, and should not, be equated to the opportunity for another person–wealthier, living in urban area– due to differences in availability of 'knowledgeable and willing' prescribers *(e.g., submissions 21, 37, 111).* In these cases, it was noted that, even after managing to get a script for medical cannabis products, timely access in rural, regional and remote location were described as being severely limited due to inadequate availability of community pharmacies to source the prescribed medical cannabis product(s). The Medicinal Cannabis Users Association of Australia submission reported that 'specialist' consultation and monitoring fees are regularly applied at 'cannabis clinics', often resulting in patients being rarely reimbursed through appropriate channels such as Medicare or other Health Fund rebates (**Submission 9; page 7**).

> "*Lack of knowledgeable or willing prescribers in rural and remote Australia have led to Australian's needing to travel to metropolitan areas to access either medical cannabis clinics or the few neurologists willing to prescribe medical cannabis. This has increased the financial burden on those families.*" (**Epilepsy Action Australia, Submission 22, page 10**)

The inconsistencies in jurisdiction-specific requirements for prescribing and/or accessing medical cannabis products have also contributed to the disparity in access to medical cannabis products. United in Compassion suggested that the creation of a 'postcode lottery', in which patients in certain States and locations have better access to medical cannabis than others have led to patients traveling and/or relocating to other States where medical cannabis is readily available, becoming 'cannabis refugees' *(Submission 7 page 4).* Stakeholders recommended harmonising and streamlining the access pathways on a national level to allow each state and territory to process applications in an equitable manner, and "ensure that geographical location is not a barrier to access treatment" *(Alcohol and Drug Foundation, Submission 26, page 3).*

> "*There should be no barriers to people wanting to legally access medicinal cannabis, whether that results from where they live or their economic status and financial well-being. Equity of access should be a fundamental guiding principle of the Commonwealth's policy settings and administrative arrangements for access to medicinal cannabis.*" (**Cancer Voices Australia, Submission 34, page 3**)

**Supply–demand mismatch.** The severe supply shortage of Australian made products, along with the fact that imported medical cannabis are not subject to Good Manufacturing Practice requirements for equivalent quality, has led to an influx of overseas products that "may or may not be of an equivalent standard to those produced under Therapeutic Goods Administration oversight in Australia" *(Entoura, Submission 25, page 4).* The supply–demand mismatch is described in the submissions as being fuelled, partly, by such reliance of imported products, which comes with problems such as stock shortages and delayed access.

> "*This past month due to a change in script requiring another approval from the TGA, and the product having to be imported from Canada, I ran out and was left without cannabis oil for almost three weeks. When I did receive the two bottles I had ordered, the expiry date was*

*within the next two months, meaning about half a bottle was due to expire before I would have been able to finish it." (**Individual; Submission 48, page. 2**).*

"*A strong domestic supply will remove the impact of a lack of commitment from, and changing priorities of, international suppliers who supply into Australia." (**Medicinal Cannabis Industry Australia, Submission 5, page. 5**).*

Epilepsy Action Australia reiterated the unique risks associated with sudden discontinuation of medical cannabis products when utilised as antiepileptic therapy due to supply issues as such disruptions in continuity of patient care places patients at "significant risk of status epilepticus and life-threatening seizures" *(**Epilepsy Action Australia, Submission 22, page. 10).***

## Affordability

Many patients referred to costs to access medical cannabis both in the process of obtaining health professional's support for and prescription to access these products and in the 'arduous' process of obtaining a supply of prescribed medical cannabis product *(e.g., see Submissions 65, 112; Committee Hansard page 2).* These include costs associated with medical reviews and appointments, accessing prescribers and/or specialised cannabis access clinics, travel and accommodation costs for rural and remote patients, and cost of filling a prescription. Patients also reported being charged by their health professionals for initial appointments or specialist consultations and for the service of having their application completed *(e.g., submission 45, 141).* The requirement of repetitive GP visits for each prescription or product substitution and reliance on imported products (i.e., additional importation and delivery fees) were described as additional costs that create a barrier to access *(e.g., Submissions 45, 70).*

"*These clinics are charging fees to put in an application to the TGA that attracts no fee. They are charging "Specialist" consultation rates and monitoring fees for which patients can rarely get a Medicare or Health Fund rebate." (**Medical Cannabis Users Assoc of Australia, Submission 9, page 7**).*

"*It cost me $200 for my initial appointment, $59 for any subsequent scripts, $80 follow up appt, $59 whenever I have to adjust dose or product, which I was able to afford by making a debt with Centrelink [sic]" (**Individual Submission 9, page. 8**).*

"*At the higher end, the delivery fee may represent over 30% of the total cost of the order while some wholesalers may not charge a delivery fee if a large quantity of products is requested through a single order. Other wholesalers charge a Dangerous Drug (DD) handling fee to offset some of the costs associated with the special storage, delivery and inventory recording requirements of these products." (**Pharmaceutical Society of Australia, Submission 16, page. 6**).*

At the time this parliamentary inquiry was called (November 2019), nabiximols was the only medical cannabis product listed in Australian Register of Therapeutic Goods (ARTG), and there was no medical cannabis product included in the PBS. Nonetheless, submissions from different stakeholders tended to support the PBS as the most appropriate way to subsidise access to medical cannabis products *(e.g., see Submissions 7, 15, 28; 29, 33, 35).* Many patients, most of whom lives with chronic medical conditions and do not have a source of income other than pensions or disability allowances, reported that they often had to choose between paying for prescribed medical cannabis and other necessities such as food and rent due to the prohibitive cost of medical cannabis products.

"*Every day we see patients crying out for cheaper or subsidised products.*" (**Medicinal Cannabis Industry Australia, Submission 5, page. 15**)

"*There is an overwhelming sense and expression of frustration and anger that there is a medicine that potentially can alleviate seizures that is only attainable by those who have the financial means to do so.*" (**Epilepsy Action Australia, submission 22, page 10**)

The subsidisation of a health professional appointment through the introduction of a new Medicare Benefits Scheme (MBS) code that considers the longer consultation time required for patient workups and the submission of medicinal cannabis applications was recommended as a solution to alleviate the 'exorbitant' *(see Submissions 8, 44, 81, 94)* cost that patients would otherwise pay at cannabis access clinics *(see Submission 21 page 22, and Submission 146 page 2).* Some stakeholders discussed that although most States and territories have an established compassionate access scheme, it is often limited to only certain patient groups with certain conditions and pointed to the need for a Federal government compassionate subsidy scheme, as an alternative to PBS, to reduce the disparity and provide equitable access to all patients.

"*A temporary subsidy [be] made available to people with epilepsy who are prescribed pharmaceutical grade cannabinoid-based medicines, until these medicines are listed on the ARTG and the PBS.*" (**Epilepsy Action Australia, Submission 22, page. 12**)

However, such blanket government subsidy models were described by some as 'unnecessary and wasteful' *(Pharmacy Guild of Australia, Submission 27, page. 3; Clinical Oncology Society of Australia, Submission 124, page. 2.)* and would potentially create a "precedent that may be used by advocates for the use of other unevaluated drugs to demand a similar subsidy from state and Federal governments solely because some patients claimed to benefit from using them" (**Submission 68, pages 7–8.**). A few submissions from patients and medicinal cannabis industry recommended that the cost of medical cannabis products could be subsidised through private health insurance, as with other non-PBS medicines *(Medicinal Cannabis Industry Australia, Submission 5, page. 4; Individual Submission 61, page. 22)*, however, such arrangements would be applicable only for non-PBS medicines which have been approved and registered in the ARTG. The Australasian College of Nutritional and Environmental Medicine (ACNEM) recommended that down-scheduling of cannabidiol would reduce the cost for pharmacies (and thus for patients) as it removes the special storage, delivery and inventory recording requirements of medical cannabis products *(ACNEM, Submission 29, page. 9.)*. The Department of Veterans Affairs (DVA) mentioned in their submissions that DVA subsidises most of the costs of medical cannabis (including consultations fees) for eligible veterans via the Repatriation Pharmaceutical Benefits Scheme, which does not have the same criteria as the PBS for listing medicines.

## Acceptability

**'Pot doctors' and 'pot heads': Stigma associated with medical cannabis.** The stigma and prejudice associated with cannabis was described by stakeholders as the main challenge for health professionals to openly discuss and/or proactively prescribe medical cannabis products, and for patients to access and use these products *(e.g., see Submissions 3, 6, 7, 22, 24, 26, 62).*

"*There is also the matter of stigma, again born out of ignorance and the assumption that anyone seeking to use Cannabis medically is a 'druggie'.*" (**Individual; submission number 76, page 6**)

Education and targeted public awareness campaigns for both healthcare consumers and professionals were also recommended as a solution to 'demystify' medical cannabis and the stigma associated with its use. In addition, providing best practice guidelines and continuous professional development courses for health professionals, revising of curriculum of medical, nursing and pharmacy schools to include endocannabinoid system, and setting up an independent national comprehensive education programme were discussed as potentials solution to the desire for further information on the use of, and access to medical cannabis products *(e.g., see submissions 3, 7, 12, 25, 27, 33, 44)*

"*Health professionals need pragmatic clinical and practical guidance so that they are well equipped to deal with patient requests to access Medicinal Cannabis in a sensitive manner, drawing upon the evidence base.*" (**Individual; Submission number 27; page 1**).

## Discussion

This thematic study examined the views and policy positions of multiple stakeholders regarding use of cannabis for therapeutic purposes, and barriers to and solutions for addressing medical cannabis access in Australia. It highlights the diverse and often conflicting stakeholder discourses around what should qualify as 'evidence', and what is safe and equitable in terms of the policy and practical frameworks for the therapeutic use of cannabis. As a morally and legally contested area, navigating the regulatory frameworks governing access to and use of cannabis-based medicines has proven to be a complex and multi-layered undertaking for end-users. In particular, the study describes the key points of tension in 'system' level medical cannabis policy and access. Findings from this study revealed that the view of government agencies and public health communities (professional health bodies and charities) in terms of the appropriateness of the current regulatory frameworks and access pathways are in strong contrast to and often incompatible with, the end-user expectations of patients, advocacy groups, and the medical cannabis industry. Submissions from government agencies and professional health bodies consistently supported maintaining the current regulatory frameworks and access pathways, whereas an overwhelming majority of patients, advocacy groups and medical cannabis industry described the current regulatory and access models as 'not fit for purpose'. Our analysis revealed that these tensions were largely underpinned by empirical uncertainties and differing views on the acceptable level and credibility of evidence i.e., what, and how much evidence is needed for policy and practice. Notwithstanding these differing views, there were commonalities among some stakeholders in terms of the various supply, regulatory, legislative, financial, and dispensing challenges that hindered timely access to medical cannabis products.

There have been several regulatory shifts in relation to medical cannabis since 2016. In December 2020, Australia voted at the Commission on Narcotic Drugs for removal of cannabis and cannabis resin from Schedule IV of the Single Convention on Narcotic Drugs, 1961, and thereby recognising the use of cannabis for therapeutic purposes. Australia was one of only three countries to support all five recommendations submitted to vote by the World Health Organisation at the commission [16, 17]. Border controls for Delta-9-tetrahydrocannabinol (dronabinol) formulations have been maintained, consistent with the current Schedule 8 poison status of cannabis that was implemented in Australia in 2017 [18]. The SAS online portal system was made available in July 2018 facilitating a concurrent application to the Commonwealth and relevant State/Territory with the aim of streamlining the application process and reducing approval times from 'several weeks to a maximum of 48 hours' [19]. The most notable regulatory changes happened in response to issues raised in the 2019 Senate

Inquiry. The Senate Community Affairs References Committee into current barriers to patient access to medicinal cannabis in Australia put forward 20 recommendations [19] including, among others

- Development and delivery of targeted education to health professionals, medical students, patients, and the public,

- Assessment of the need for establishing a new independent medical cannabis regulator, and a Commonwealth Compassionate Access Subsidy Scheme

- Amendment and/or clarification around the requirements of SAS-B and AP schemes, and

- Down-scheduling or de-scheduling of low dose cannabidiol.

The Australian Government published a response on the 4 March 2021 to the recommendations arising from the Senate Inquiry. Details on which recommendations were noted, accepted in part or in full, not accepted can be found at the department's website [19]. The government have since made a series of reforms to the way medical cannabis is accessed via SAS-B and Authorised Prescriber schemes as well as provided tailored information for consumers, health professionals, sponsors and manufacturers. Another major regulatory change came into effect on 1 February 2021, when cannabidiol at a dose of 150 mg or less and meeting specific requirements was down-scheduled to be available over the counter [20], albeit no product has met requirements to be supplied in this manner yet. According to the Therapeutic Goods Administration Medicinal cannabis Special Access Scheme Category B dashboard data, there has been a surge in the number of SAS-B applications, increasing from 57,700 applications in 2020 to 111,500 in 2021 (as of November 2021) [21]. On the 22 November 2021, the Therapeutic Goods Administration implemented changes to the SAS and AP application pathways whereby applications are made by cannabinoid content categories. However, it is not yet apparent how bioequivalence will be determined in this instance. This number is expected to increase substantially given the recent changes to SAS-B application requirements.

Submissions to the Senate Inquiry revealed a broad range of factors that may contribute to inequity in patient access to medical cannabis, although it is worth noting that many of these factors also influence access to healthcare more broadly. These include: a lack of regulatory harmonisation across states and territories; medical practitioner workforce disparities and limited access to health professionals willing to prescribe cannabis medicines; inequalities relating to participation in clinical trials; public transport inequities; and costs associated with medical cannabis access. There is disparity in the number of SAS-B applications approved in each state and territory, where the number of applications approved in Queensland is almost twice that of New South Wales, well over double that of Victoria, and six times that of Western Australia [21]. Approvals for other states and territories of Australia including South Australia, Australian Capital Territory, Northern Territory, and Tasmania are negligible [21]. It is, however, important to note that the SAS B approval data represents the prescriber consulting location rather than the patient location. For example, a product may be prescribed following a tele-health consultation by a prescriber in one state for a patient residing in another state and the product can be shipped to patients from a dispensary located in a different state or territory. The low rate of applications from Australian Capital Territory is likely attributable to the state's decriminalisation of cannabis for personal use since 31 January 2020 [22]. Currently, the majority of clinical trials that provide subsided medical cannabis to patients are centred around trials that are in the states of New South Wales, Queensland and Victoria [23]. Inequities are also compounded by the geographic location of cannabis clinics in these same states [24]. Disparities are no more evident than they are in Tasmania, which has the lowest number

of approvals in Australia due to the latent introduction of legislation to allow both general practitioners and specialists to prescribe, requirement that both prescribers and dispensers need to be from Tasmania, and legislation that did not allow prescribers to access the national Therapeutic Goods Administration streamlined SAS-B portal for prescribing [25, 26] There are also issues around access in rural communities within each state and territory where limited and variable medical workforce supply in rural districts, poor access to community pharmacies to source the prescribed medical cannabis product(s), and conflicting opinions on current drug driving laws prohibiting driving [25]. These leads to heightened financial strain and required resources to access medical cannabis products in geographical locations that have lower socioeconomic status compared to urban areas.

Acceptability of medical cannabis to prescribers and the public has also been reported by diverse stakeholders as a major psychosocial factor that limits patient access to medical cannabis. Despite a growing interest in and awareness of its benefits, several stigmas remain attached to medical cannabis, such as its connections to cannabis more broadly, including negative perceptions as a recreational substance, illegality, and a potential vulnerability to or exacerbation of mental health concerns [27]. These stigmas prevent both consumers and clinicians from openly discussing and engaging with medical cannabis as a treatment possibility. Given that these stigmas toward therapeutic use of cannabis are mainly due to its connections to recreational cannabis, it may partially be addressed by decoupling recreational vs medical cannabis via increased education and public awareness campaigns, as recommended by the Senate Inquiry committee. While concerns regarding the acceptability of medical cannabis may equally be viewed as attempting to ensure patient safety, they conversely result in making access to medical cannabis more difficult for consumers. Evidence strongly suggests that the complex regulatory framework for accessing medical cannabis is contributing to some people choosing to access illicit cannabis products, which has its own quality and safety implications and legal repercussions.

The financial and temporal costs associated with accessing medical cannabis, incurred from accessing appointments and products, have increased exponentially and in many cases is viewed as prohibitive. Patients must pay for all medical cannabis products outright; except for Epidyolex® which was listed on the Pharmaceutical Benefits Scheme in May 2021 as a third line treatment for Dravet Syndrome [6, 28]. The prohibitive cost and complex regulatory landscape were indicated by consumers as main reasons to resort to illicit markets and/or self-cultivation for sourcing cannabis products for therapeutic purposes. Although the recommendation response report [19] details that 'the Government does not support the provisions of amnesties for possession and/or cultivation of cannabis through illegal sources', a considerable proportion of consumers reported having used illicit markets or self-cultivation routes to access their products. This was also reported in a national survey conducted in 2018–19, where more than 90% of respondents indicated they obtained their medical cannabis from 'recreational dealers', illicit medical cannabis suppliers, friends or family, or via self-cultivation [29].

Internationally, the question of how to regulate medical cannabis is a major source of debate in many countries and continues to divide the public health community. The laws and policy domains in the majority of countries are grounded in and shaped by the existing and historical cannabis regulation [30], making it difficult to draw credible comparisons across countries. Despite differences in their regulatory approaches to medical cannabis, majority of these countries agree on: i) the need for continued education for health practitioners, ii) addressing stigma, iii) involving patients and addressing patients' concerns, and iv) improving the evidence base through real-world data collection.

The 2019–20 Senate Inquiry and transcripts from public hearings represent the most recent and comprehensive data from multiple stakeholders on use of, and access to medical cannabis in Australia. However, our study has some limitations. The major caveat with respect to the dataset is the fact that submissions to the inquiry were voluntary and that the data may not be nationally representative. Submissions by public health academics and researchers were made in their individual capacity and views of professional health bodies may not necessarily represent the views of all their members. It is also worth mentioning that it was unclear in the submissions whether respondents are of Aboriginal or Torres Strait Islander backgrounds. There is a need for more research to better understand the values and preferences of diverse communities. The dataset and findings from the analysis should be regarded as a snapshot in time, given the ever-changing regulatory landscape regarding medical cannabis. As discussed in detail, some of the 'system-level' access barriers have been resolved in the some of the reforms and regulatory changes that were instituted since the inquiry was held. Nevertheless, this analysis has provided an insight into the views of multiple stakeholders on use of medical cannabis and the Australian regulator approach, and achieving equitable access to these products.

## Conclusions

Our analysis of Inquiry submissions revealed various regulatory, financial, psychological, supply and dispensing challenges with potential impact on timely access to medical cannabis products. While recent regulatory changes made by the jurisdictional and Federal government have been implemented with the intention of addressing perceived access barriers to medical cannabis, the Senate Inquiry highlighted broader access issues relevant to all medicines. One of the fundamental barriers that determine how and if a patient accesses medicines is the urban–rural disparity in access to healthcare. Furthermore, decoupling recreational vs medical cannabis and reframing societal and health professional's views of medical cannabis are some of essential first steps to destigmatise and improve access to the therapeutic use of cannabis.

## Supporting information

**S1 File. List of stakeholders included in the analysis (n = 121).**
(DOCX)

## Author Contributions

**Conceptualization:** Daniel Erku, Lisa-Marie Greenwood, Myfanwy Graham, Christine Mary Hallinan, Jessica G. Bartschi, Elianne Renaud, Paul Scuffham.

**Data curation:** Daniel Erku, Jessica G. Bartschi.

**Formal analysis:** Daniel Erku, Lisa-Marie Greenwood, Myfanwy Graham, Christine Mary Hallinan, Jessica G. Bartschi, Paul Scuffham.

**Methodology:** Daniel Erku, Christine Mary Hallinan.

**Supervision:** Paul Scuffham.

**Writing – original draft:** Daniel Erku, Lisa-Marie Greenwood, Myfanwy Graham, Christine Mary Hallinan, Elianne Renaud, Paul Scuffham.

**Writing – review & editing:** Daniel Erku, Lisa-Marie Greenwood, Myfanwy Graham, Christine Mary Hallinan, Jessica G. Bartschi, Elianne Renaud, Paul Scuffham.

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
