## [Decision Letter · Decision Letter 0]

2 Sep 2022

PONE-D-22-15976From growers to patients: multi-stakeholder views on the use of, and access to medicinal cannabis in AustraliaPLOS ONE

Dear Dr. Erku,

Thank you for submitting your manuscript to PLOS ONE. After careful consideration, we feel that it has merit but does not fully meet PLOS ONE’s publication criteria as it currently stands. Therefore, we invite you to submit a revised version of the manuscript that addresses the points raised during the review process. Your manuscript has been reviewed by two peer-reviewers and their comments are appended below.  The reviewers have commented that some of the terminology used in the study could be better explained or clarified, and indicated some aspects of the study that could be strengthened with further discussion.  Can you please address their comments?

We look forward to receiving your revised manuscript.

Kind regards,

Maria Elisabeth Johanna Zalm, Ph.D

Editorial Office

PLOS ONE

Journal Requirements:

a) Did participants provide their written or verbal informed consent to participate in this study?

Reviewers' comments:

Reviewer's Responses to Questions

**Comments to the Author**

1. Is the manuscript technically sound, and do the data support the conclusions?

Reviewer #1: Yes

Reviewer #2: Yes

2. Has the statistical analysis been performed appropriately and rigorously? 

Reviewer #1: Yes

Reviewer #2: N/A

3. Have the authors made all data underlying the findings in their manuscript fully available?

Reviewer #1: Yes

Reviewer #2: Yes

4. Is the manuscript presented in an intelligible fashion and written in standard English?

Reviewer #1: Yes

Reviewer #2: Yes

5. Review Comments to the Author

Reviewer #1: Overall a very good paper, providing interesting insights and perspectives into the Senate discussions and their outcome. I only missed some considerations of possible traditional and herbal medicine schemes that exist in the country or in states and territories, and may have been used as a point of comparison (see PP. 157-160 in WHO global report on traditional and complementary medicine 2019, https://www.who.int/publications/i/item/978924151536). A mention and brief discussion of the participation (or lack thereof) of aboriginal people would also have been welcomed.

Some comments:

P.1: "Community interest in the use of cannabis for medicinal purposes has increased within Australia and internationally [1, 2]. The rapid growth in national demand for access to cannabis for therapeutic use has largely been driven by..."

I would suggest to harmonize the language you use elsewhere in the article, and general use: "therapeutic" or "medical" to be applied to purposes (therapeutic purposes, therapeutic use...), and the word "medicinal" to the plant (medicinal plants; medicinal cannabis). To match this, the sentence could therefore read as follows:

"Community interest in the use of cannabis for *therapeutic* purposes has increased within Australia and internationally [1, 2]. The rapid growth in national demand for access to *medicinal* cannabis for *such* use has largely been driven by..."

--

P.1: "consumer advocacy groups"

Consider "patient advocacy groups" or "patient or consumer advocacy groups"

--

P. 1, § 2: "(like many international jurisdictions)"

Consider: "(like many other jurisdictions internationally)" or "(like many other jurisdictions worldwide)"

--

P. 4, in the sentence "Between 17 January 2020 and 26 March 2020, multiple stakeholders, including consumers, clinicians, and organisations, submitted their perspectives relating to the terms of reference."

Consider using "patients" instead of "consumers" – Note: this is valid elsewhere, throughout the text of the manuscript.

I would also recommend replacing "Between 17 January 2020 and 26 March 2020" by "During a XX-days consultation in 2020". My assumption is that such level of precision is not required in the introduction; however, the mention of exact dates in the "Data sources" part of the "Methods" section where other dates (e.g. December 2019; January 2020) are mentioned, may make more sense.

--

P. 3 or 4:

It may be relevant to mention that Australia was one of the three countries that voter in favour of all the recommendations of the World Health Organization related to medicinal cannabis, in December 2020. See table p.7 in: History, science, and politics of international cannabis scheduling, 2015–2021 https://ssrn.com/abstract=3932639.

Explanations of vote pp. 17-18 in: https://www.unodc.org/documents/commissions/CND/CND_Sessions/CND_63Reconvened/ECN72020_CRP24_V2007524.pdf

Following the vote, Australian representatives declared: "we must be prepared to listen to expert scientific and medical advice and keep the scheduling of controlled substances up to date and in line with community expectations" and that "Australia will continue to advocate for greater access to controlled medicines for those in need" https://www.unodc.org/documents/commissions/CND/CND_Sessions/CND_64/Statements/13April/Australia.pdf

--

P. 5: "government and non-government organisations, peak health bodies, health charities, advocacy groups, medicinal cannabis industry and members of the public"

I have some issues with the classification of stakeholders presented here. And, although the manuscript points at the Supplementary File 1, there is no further explanation of which stakeholder was classified in which particular category, and the reasons for it. In my understanding, the list could be reduced to three classes: "government organisation", "non-government organisation" (which includes de facto "health charities, advocacy groups, medicinal cannabis industry" -unsure about "peak health bodies") and "members of the public" (i.e. individuals).

I am not sure of what "peak health bodies" refer to, although I understand I may not be familiar with this terminology. There are some examples throughout the text which help get an idea, but it would be interesting to explain more clearly whether these include also public or semi-public health monitoring agencies, institutions, and other public health organisms of a public-interest character, or if it refers only to private health stakeholders, consortia, or colleges?

I would also have appreciated a clarification of the criteria used to distinguish between "advocacy groups" and "health charities" –my understanding is that many of those labelled as one group could fall under the other, and vice-versa. E.g.: are "Drug Free Queensland" and "Medical Cannabis Users Assocation of Tasmania" advocacy groups, health charities... or both?

Finally, it may also have been beneficial to screen "members of the public" (1) to distinguish healthcare professionals or patients, for instance, from other interested parties (2) for possible affiliation to non-governmental organisations and/or possible corporate interest.

In addition to clarifications on P.5 (maybe a small table describing characteristics of the different classes of stakeholders), an idea could be to present the information in Supplementary File 1 as a table, with a first column listing stakeholders, and added columns with precisions about the class applied by the authors to each particular respondent, and other comments or information related. –in a way that matches the information presented in the first paragraph of the "Results" section.

P. 6 mentions the presence of "a link to included documents" in the Supplementary File, but there was no link(s). It may be interesting to provide URLs to the submissions, if available.

--

P. 6: "Submissions were also supplemented with interview transcripts from the public hearing (Table 1)."

No Table 1 in the manuscript I have been given access to.

--

P. 7: "Submissions from the health and medical field" is followed by a quote from GW pharmaceutical, supporting the exact type of products that this company is commercialising. Tis does not seem to be just any submission from simply "the health and medical field", but from a multinational corporate firm with interest in preserving market shares. This should be acknowledged, notably in the context of the two only cannabis-based medicines listed on the ARTG, whiich are mentioned P. 3, are owned and commercialised by that particular company. This seems to need acknowledgement as well.

This is echoed on P. 9, where the sentence "Submissions from some pharmaceutical industry entities with an interest in the therapeutic use of cannabis have framed ‘medicinal cannabis’ as consumer products that lacked sufficient empirical evidence to be considered an approved medicine." could be more precise, for instance:

"Submissions from some pharmaceutical industry entities with *a commercial* interest in the therapeutic use of *particular products of*/*their products* cannabis have framed ‘medicinal cannabis’ as consumer products that lacked sufficient empirical evidence to be considered an approved medicine."

--

P. 8: "Whilst the need for high quality evidence on the safety and efficacy of medicinal cannabis products was widely acknowledged, there was a sense of recognition that there needs to be a way for medicinal cannabis to be managed and accessible to patients (even if conditionally) within the therapeutic context."

Unclear sentence: was there a *shared* sense of recognition among all stakeholders or only some? It may need to be specified.

--

P. 11 where "tetrahydrocannabinol (THC)" is mentioned, it could be sound to include the international nonproprietary name (INN) for delta-9-tetrahydrocannabinol, which is "dronabinol". This would make sense, provided that the INN for Epidiolex® is used on P. 3 (INN cannabidiol) as well as the non-proprietary name for Sativex® (nabiximols) PP. 3 & 15.

See https://www.linkedin.com/pulse/dronabinol-your-cannabis-kenzi-riboulet-zemouli/ and P. 6 in: List of Proposed INNs No. 51, WHO Chronicle 38(2), 1984 https://cdn.who.int/media/docs/default-source/international-nonproprietary-names-(inn)/pl51.pdf?sfvrsn=3ded55af_9&download=true and P. 4 in: List of Recommended INNs No. 24, WHO Chronicle 38(6), 1984 https://cdn.who.int/media/docs/default-source/international-nonproprietary-names-(inn)/rl24.pdf?sfvrsn=99779214_6&download=true

--

P. 11 "Under the current Australian drug driving policy, driving while having detectable levels of tetrahydrocannabinol (THC) in the body is considered an offence, and patients may be subject to prosecution, fines, and loss of licence."

Source or reference needed.

--

P. 11 "A significant proportion of patients who made submissions reported that they resorted to black market (also referred to as 'black', 'grey' or 'green' market) to source their medicinal cannabis products"

Consider: "they resorted to *the illicit* market (also referred to as 'black', 'grey', *'alternative',* or 'green' market)"

--

P. 11 " reported self-producing medicinal cannabis for personal use, often without a license to produce medicinal cannabis (see submissions 13; 80, 91, 29; 4, 110, 44, 6)."

Consider " reported self-producing medicinal cannabis for personal use, often without a license to *that purpose*"

Also consider ordering the submission numbers in ascending order, in this sentence and throughout the text.

--

P. 12: "many patients who opted to self-medicate with illicit cannabis"

Self-medication is different than sourcing cannabis from the illicit market after being prescribed. Self-medication implies the absence of an external healthcare advice. I suggest replacing the terms " who opted to self-medicate with illicit cannabis" with "who opted to source their cannabis from the illicit market"

--

P. 12: "issues associated with sourcing medicinal cannabis products from the black market"

Consider relying on neutral language, except in quotations, i.e. using the terms "illicit market" instead of "black market" throughout the text (e.g. P. 20).

--

PP. 13 & 14, sentences almost similar:

"The Medicinal Cannabis Users Association of Australia submission reported that where ‘cannabis clinics’ are available, often via telehealth consultation, they often charge "specialist" consultation rates and monitoring fees for which patients can rarely get a Medicare or Health Fund rebate (submission 9; page 7)."

"“These clinics are charging fees to put in an application to the TGA that attracts no fee. They are charging "Specialist" consultation rates and monitoring fees for which patients can rarely get a Medicare or Health Fund rebate.” (Medical Cannabis Users Assoc of Australia, Submission 9, page 7)."

--

P. 15: "At the time of this parliamentary inquiry (November 2019)"

I thought that it was between 17 January 2020 and 26 March 2020,?

--

P. 15: "compassionate access schemes"  "scheme"

--

P. 18: "However, Australia did not vote for additional recommendations made by the World Health Organisation, including the removal of border controls for Delta-9-tetrahydrocannabinol formulations"

This is incorrect, or I did not properly understand the formulation.

A good article to understand the WHO recommendations is WHO’s first scientific review of medicinal Cannabis: from global struggle to patient implications https://www.emerald.com/insight/content/doi/10.1108/DHS-11-2021-0060/full/html

Australia voted in favour of all recommendations that were submitted to the vote, and was one of the few countries to do so. The recommendations that Australia did not vote in favour of were, in fact, not put on the ballot. See vote results here: “Report on the reconvened sixty-third session (2-4 December 2020), economic and social council official records, 2020 supplement no. 8A”, E/2020/28/Add.1, available at: undocs.org/E/2020/28/Add.1 and in https://ssrn.com/abstract=3932639

See also explanations of vote pp. 17-18 in: https://www.unodc.org/documents/commissions/CND/CND_Sessions/CND_63Reconvened/ECN72020_CRP24_V2007524.pdf where is is stated:

"Australia supported recommendation 5.1 which recogni zes that cannabis and cannabis resin are unlikely to produce ill-effects similar to other Schedule IV substances, and that some parties have enacted laws to enable the medical use of cannabis and cannabis resin.

− Australia considers this approach strikes an appropriate regulatory balance and simply recognizes that there is a legitimate medical use quite separate to the use of cannabis for research purposes.

Australia supported recommendation 5.2 and 5.3 Australia’s strong preference is for dronabinol, THC and its stereoisomers to be managed consistently with cannabis and cannabis resin under Schedule I of the 1961 Convention. This would remove ambiguity and complexity, especially as it applies to trade between member states.

Australia supported recommendation 5.4 which will improve consistency and remove ambiguity in the regulation of cannabis extracts and preparations.

Australia supported recommendation 5.5 on the basis that predominantly cannabidiol preparations with low concentrations delta-9 THC are of less risk of harm than those which contain greater than 0.2 per cent. This recommendation assists improving consistency and trade and reporting discrepancies."

--

P. 18: "no product has met requirements to be supplied in this manner [16]"

It is unclear if the reference mentioned related to the fact that no product has met requirements to be supplied in this manner. I would place the reference before the word "albeit", and slightly edit the sentence so it reads:

"over the counter,[16] albeit no product has met requirements to be supplied in this manner *yet*."

--

P. 20: typo in "or by via self-cultivation".

--

I look very much forwards to reading an improved and final version of this article!

Thank you for your work.

Reviewer #2: A fascinating and well written article that qualitatively delves into over 120 submissions made by a variety of stakeholders to the 2019 Australian inquiry into barriers to patient access to medicinal cannabis. Using the modified Levesque patient-centred access to care framework, the authors examine these barriers from the perspective of appropriateness, availability and geographical accessibility, acceptability, and affordability. The analysis uncovers the tension and differences between government/professional and patient/advocacy stakeholders with regards to their perception of the current regulatory framework and the impact on access and use of medical cannabis. This manuscript provides a deep dive into the discourse surrounding access to medicinal cannabis in Australia and will be informative for other jurisdictions who have legalized/decriminalized access to medicinal cannabis.

There are a few issues that would warrant further consideration and elucidation:

• In the Introduction, a description of the SAS Category A and SAS-B pathways is presented – it is not clear if there is a distinction between “medical practitioners” and “registered health practitioners”. Given some jurisdictions allow other health professionals, such as nurse practitioners, to authorize/prescribe medical cannabis, it would be helpful to clarify if there is indeed a difference across these pathways re: type of health care professional.

• There is reference throughout the manuscript to “scientific evidence” and what type and how much evidence is perceived to be required by different stakeholders to support the prescription of medical cannabis. The role of patient preference/values and contextual factors, such as the risks of other pharmaceutical treatments (e.g., opioids) and resources, however, is not mentioned. Given these are hallmarks of evidence-based medicine, it would be worthwhile for a brief discussion about whether evidence-based medicine is a guiding principle across government, professional and charity groups, as well as among the patient/advocacy stakeholders.

• I was intrigued by the quote drawn from the health and medical field that suggested that “We firmly believe that patients deserve access to medicines of the highest quality – testing for both safety and efficacy in placebo-controlled, randomized clinical trials and licensed as medicines…”. This quote appears quite naïve of the reality of medical cannabis research with regards to limited funding as well as the challenges associated with conducting clinical trials on a plant-based medicine that is difficult standardize as well as blind subjects to due to its psychoactive effect. It also expresses a clear bias toward the medicalization of cannabis, assuming that medicinal cannabis should be developed into a pharmaceutical medicine that can be patented and licensed. I would encourage the authors to reflect further in the discussion on the tension that exists when these two different paradigms of therapies (plant vs. pharmaceutical) collide.

• The discussion would have benefitted from a brief discussion of other medical cannabis regulatory systems, including in Canada and the US, and whether the associated policies/processes would address some of the barriers expressed by stakeholders in Australia.

• Recommend that not only do medical and pharmacy schools need to include education on the ECS and medicinal cannabis, but also nursing schools as nurses are often tasked in clinical settings to assist patients in using medicinal cannabis. To do so, they require sufficient education about medicinal cannabis.

• In the Discussion, there is commentary regarding the acceptability of medical cannabis being viewed as an attempt to ensure patient safety; however, in restricting access and pushing patients towards the unregulated/illegal market, they are also placing them at risk through the use of an unregulated supply of cannabis as well as the risk of criminalization. Some discussion of this is required.

• The need to “decouple” medicinal cannabis from that used for recreational purposes is indeed important; however, it would be useful to suggest some strategies to do (i.e., education, public education, working with professional societies with stigmatizing policies).

Minor grammatical/writing issues:

• The term “peak health bodies” is unique to Australia and does not easily translate for readers from other countries. Please define when first used and provide an example.

• The word “neogitations” is spelled wrong on the 3rd line above the section entitled “The role of evidence in informing policy and clinical practice…”

• Remove “an” from the following sentence, “…and advocated for an expedited access while TGA registration is pending.”

• In the first sentence under “Supply – demand mismatch”, edit to read: “…Australian made products, along with the fact that imported medicinal cannabis…”

• Edit the following sentence in the discussion: “…cost that patients would otherwise pay at cannabis access clinics…”

• There are too many acronyms, particularly in the Discussion. Please consider keeping only a handful of them so the reader is not confused or searching for what they stand for throughout the manuscript.

• The sentence with “This ‘postcode’ lottery is evidence “ appears to be unfinished in the Discussion.

• The term “black market” has fallen out of favour; please consider using “unregulated” or “illegal” market instead.

6. PLOS authors have the option to publish the peer review history of their article (what does this mean?). If published, this will include your full peer review and any attached files.

Reviewer #1: **Yes: **Kenzi Riboulet-Zemouli

Reviewer #2: No

---

## [Author Response · Author response to Decision Letter 0]

14 Sep 2022

Reviewer 1:

Overall, a very good paper, providing interesting insights and perspectives into the Senate discussions and their outcome. I only missed some considerations of possible traditional and herbal medicine schemes that exist in the country or in states and territories and may have been used as a point of comparison (see PP. 157-160 in WHO global report on traditional and complementary medicine 2019, https://www.who.int/publications/i/item/978924151536). A mention and brief discussion of the participation (or lack thereof) of aboriginal people would also have been welcomed.

Thank you for this feedback. It was unclear in the submissions whether respondents are of Aboriginal or Torres Strait Islander backgrounds. We have indicated this as a limitation, and recommended for more research to better understand the values and preferences of diverse communities.

1. P.1: "Community interest in the use of cannabis for medicinal purposes has increased within Australia and internationally [1, 2]. The rapid growth in national demand for access to cannabis for therapeutic use has largely been driven by..."

I would suggest to harmonize the language you use elsewhere in the article, and general use: "therapeutic" or "medical" to be applied to purposes (therapeutic purposes, therapeutic use...), and the word "medicinal" to the plant (medicinal plants; medicinal cannabis). To match this, the sentence could therefore read as follows:

"Community interest in the use of cannabis for *therapeutic* purposes has increased within Australia and internationally [1, 2]. The rapid growth in national demand for access to *medicinal* cannabis for *such* use has largely been driven by..."

This has been changed (pg. 3 and pg. 18), text copied below: 

Pg. 3: “Community interest in the use of cannabis for medicinal therapeutic purposes has increased within Australia and internationally [1, 2].”

Pg: 18: “…recognising the use of cannabis for medicinal therapeutic purposes.”

2. P.3: "Consumer advocacy groups" - consider "patient advocacy groups" or "patient or consumer advocacy groups".

This has been changed (pg. 3 and pg. 9, text copied below):

“The rapid growth in national demand for access to cannabis for therapeutic use has largely been driven by patient or consumer advocacy groups…”

“Again, clinical and governmental perspectives were not typically reflective of the views of patients and patient or consumer advocacy groups…”

3. P. 3: "(like many international jurisdictions)" - consider "(like many other jurisdictions internationally)" or "(like many other jurisdictions worldwide)".

This has been changed (pg. 3, text copied below):

“Alongside these legislative changes, Australia (like many other jurisdictions worldwide)…”

4. P. 4, In the sentence "Between 17 January 2020 and 26 March 2020, multiple stakeholders, including consumers, clinicians, and organisations, submitted their perspectives relating to the terms of reference", consider using "patients" instead of "consumers" – Note: this is valid elsewhere, throughout the text of the manuscript. I would also recommend replacing "Between 17 January 2020 and 26 March 2020" by "During a XX-days consultation in 2020". My assumption is that such level of precision is not required in the introduction; however, the mention of exact dates in the "Data sources" part of the "Methods" section where other dates (e.g. December 2019; January 2020) are mentioned, may make more sense.

This has been changed (pg. 4, text copied below):

“During 69 days of consultation in 2020, multiple stakeholders, including patients, clinicians, and organisations, submitted their perspectives relating to the terms of reference…”

5. P. 3 or 4: It may be relevant to mention that Australia was one of the three countries that voter in favour of all the recommendations of the World Health Organization related to medicinal cannabis, in December 2020. See table p.7 in: History, science, and politics of international cannabis scheduling, 2015–2021 https://ssrn.com/abstract=3932639. Explanations of vote pp. 17-18 in: https://www.unodc.org/documents/commissions/CND/CND_Sessions/CND_63Reconvened/ECN72020_CRP24_V2007524.pdf Following the vote, Australian representatives declared: "we must be prepared to listen to expert scientific and medical advice and keep the scheduling of controlled substances up to date and in line with community expectations" and that "Australia will continue to advocate for greater access to controlled medicines for those in need" https://www.unodc.org/documents/commissions/CND/CND_Sessions/CND_64/Statements/13April/Australia.pdf.

Thank you to the reviewer for this clarification. We have included this information as part of the discussion on regulatory changes. We have included the suggested citations. See the extract below. 

p.g. 18 “In December 2020, Australia voted at the Commission on Narcotic Drugs for removal of cannabis and cannabis resin from Schedule IV of the Single Convention on Narcotic Drugs, 1961, and thereby recognising the use of cannabis for therapeutic purposes. Australia was one of only three countries to support all five recommendations submitted to vote by the World Health Organisation at the commission. Border controls for Delta-9-tetrahydrocannabinol (dronabinol) formulations have been maintained, consistent with the current Schedule 8 poison status of cannabis that was implemented in Australia in 2017 [14].”

6. P. 5: "government and non-government organisations, peak health bodies, health charities, advocacy groups, medicinal cannabis industry and members of the public"

I have some issues with the classification of stakeholders presented here. And, although the manuscript points at the Supplementary File 1, there is no further explanation of which stakeholder was classified in which particular category, and the reasons for it. In my understanding, the list could be reduced to three classes: "government organisation", "non-government organisation" (which includes de facto "health charities, advocacy groups, medicinal cannabis industry" -unsure about "peak health bodies") and "members of the public" (i.e. individuals).

I am not sure of what "peak health bodies" refer to, although I understand I may not be familiar with this terminology. There are some examples throughout the text which help get an idea, but it would be interesting to explain more clearly whether these include also public or semi-public health monitoring agencies, institutions, and other public health organisms of a public-interest character, or if it refers only to private health stakeholders, consortia, or colleges?

I would also have appreciated a clarification of the criteria used to distinguish between "advocacy groups" and "health charities" –my understanding is that many of those labelled as one group could fall under the other, and vice-versa. E.g.: are "Drug Free Queensland" and "Medical Cannabis Users Assocation of Tasmania" advocacy groups, health charities... or both?

Finally, it may also have been beneficial to screen "members of the public" (1) to distinguish healthcare professionals or patients, for instance, from other interested parties (2) for possible affiliation to non-governmental organisations and/or possible corporate interest.

7. In addition to clarifications on P.5 (maybe a small table describing characteristics of the different classes of stakeholders), an idea could be to present the information in Supplementary File 1 as a table, with a first column listing stakeholders, and added columns with precisions about the class applied by the authors to each particular respondent, and other comments or information related. –in a way that matches the information presented in the first paragraph of the "Results" section.

8. P. 6 mentions the presence of "a link to included documents" in the Supplementary File, but there was no link(s). It may be interesting to provide URLs to the submissions, if available. Also, "Submissions were also supplemented with interview transcripts from the public hearing (Table 1).” - no Table 1 in the manuscript I have been given access to.

Thank you. We have now included i) table 1 describing characteristics of the different classes of stakeholders, and ii) a supplementary table with list of included submissions and category of stakeholders. We have also included a link to website link to the original submissions publicly available on the Australian government parliament website

9. P. 7: "Submissions from the health and medical field" is followed by a quote from GW pharmaceutical, supporting the exact type of products that this company is commercialising. Tis does not seem to be just any submission from simply "the health and medical field", but from a multinational corporate firm with interest in preserving market shares. This should be acknowledged, notably in the context of the two only cannabis-based medicines listed on the ARTG, whiich are mentioned P. 3, are owned and commercialised by that particular company. This seems to need acknowledgement as well. This is echoed on P. 9, where the sentence "Submissions from some pharmaceutical industry entities with an interest in the therapeutic use of cannabis have framed ‘medicinal cannabis’ as consumer products that lacked sufficient empirical evidence to be considered an approved medicine." could be more precise, for instance:

"Submissions from some pharmaceutical industry entities with *a commercial* interest in the therapeutic use of *particular products of*/*their products* cannabis have framed ‘medicinal cannabis’ as consumer products that lacked sufficient empirical evidence to be considered an approved medicine."

We echo the reviewer’s comment that a submission from cannabis/pharmaceutical industry do not necessarily reflect the views of “health and medical field”. We have therefore replaced the quote from GQ Pharmaceuticals to a representative quote from a health and medical profession. See the quote below. 

“The gaps are substantial in current knowledge about the dose, delivery of different products, therapeutic use as add-on therapy or stand-alone therapy in the treatment of a broad spectrum of conditions and diseases. This poses an unacceptable risk in my view to changing the current requirements of registration that have a remarkable track record.” (Professor James Angus, Chair of the Australian Advisory Council on the Medicinal Use of Cannabis’ Submission 53; page 2)

We have also made the recommended changes as “Submissions from some pharmaceutical industry entities with a commercial interest in the therapeutic use of particular products of cannabis (e.g., cannabis products registered in ARTG) have framed ‘medicinal cannabis’ as consumer products that lacked sufficient empirical evidence to be considered an approved medicine.”

10. P. 8: "Whilst the need for high quality evidence on the safety and efficacy of medicinal cannabis products was widely acknowledged, there was a sense of recognition that there needs to be a way for medicinal cannabis to be managed and accessible to patients (even if conditionally) within the therapeutic context.” - unclear sentence: was there a *shared* sense of recognition among all stakeholders or only some? It may need to be specified.

This has been clarified (pg. 8, text copied below):

“…there was a shared sense of recognition by stakeholders that there needs to be a way for medicinal cannabis to be managed and accessible to patients (even if conditionally)…”

11. P. 11 where "tetrahydrocannabinol (THC)" is mentioned, it could be sound to include the international nonproprietary name (INN) for delta-9-tetrahydrocannabinol, which is "dronabinol". This would make sense, provided that the INN for Epidiolex® is used on P. 3 (INN cannabidiol) as well as the non-proprietary name for Sativex® (nabiximols) PP. 3 & 15. See https://www.linkedin.com/pulse/dronabinol-your-cannabis-kenzi-riboulet-zemouli/ and P. 6 in: List of Proposed INNs No. 51, WHO Chronicle 38(2), 1984 https://cdn.who.int/media/docs/default-source/international-nonproprietary-names-(inn)/pl51.pdf?sfvrsn=3ded55af_9&download=true and P. 4 in: List of Recommended INNs No. 24, WHO Chronicle 38(6), 1984 https://cdn.who.int/media/docs/default-source/international-nonproprietary-names-(inn)/rl24.pdf?sfvrsn=99779214_6&download=true

This has been included (pg. 11 and pg. 18, text copied below):

“… detectable levels of tetrahydrocannabinol (THC; dronabinol) in the body…”

“… border controls for Delta-9-tetrahydrocannabinol (dronabinol) formulations…”

12. P. 11 "Under the current Australian drug driving policy, driving while having detectable levels of tetrahydrocannabinol (THC) in the body is considered an offence, and patients may be subject to prosecution, fines, and loss of licence."

Source or reference needed.

We have included the following references. 

• Perkins D, Brophy H, McGregor IS, O'Brien P, Quilter J, McNamara L, Sarris J, Stevenson M, Gleeson P, Sinclair J, Dietze P. Medicinal cannabis and driving: the intersection of health and road safety policy. Int J Drug Policy. 2021 Nov;97:103307 

• Alcohol and Drug Foundation. Medical Cannabis and Driving in Australia. 2022

13. P. 11 "A significant proportion of patients who made submissions reported that they resorted to black market (also referred to as 'black', 'grey' or 'green' market) to source their medicinal cannabis products" - consider: "they resorted to *the illicit* market (also referred to as 'black', 'grey', *'alternative',* or 'green' market)"

This has been changed (pg. 11, text copied below):

“A significant proportion of patients who made submissions reported that they resorted to illicit markets (also referred to as 'black', 'grey' or 'green' market) to source their medicinal cannabis products…”

14. P. 11 " reported self-producing medicinal cannabis for personal use, often without a license to produce medicinal cannabis (see submissions 13; 80, 91, 29; 4, 110, 44, 6)” - consider " reported self-producing medicinal cannabis for personal use, often without a license to *that purpose*". Also consider ordering the submission numbers in ascending order, in this sentence and throughout the text.

This has been changed (pg. 11, text copied below). Submission numbers have now been placed in ascending order where appropriate throughout the article.

“…reported self-producing medicinal cannabis for personal use, often without a license to produce medicinal cannabis for that purpose (see submissions 4, 6, 13, 29, 44, 80, 91, 110).”

15. P. 12: "many patients who opted to self-medicate with illicit cannabis" - self-medication is different than sourcing cannabis from the illicit market after being prescribed. Self-medication implies the absence of an external healthcare advice. I suggest replacing the terms " who opted to self-medicate with illicit cannabis" with "who opted to source their cannabis from the illicit market"

This has been changed (pg. 12, text copied below):

“However, many patients who opted to source their cannabis from illicit markets felt that they had no other choice…”

16. P. 12: "issues associated with sourcing medicinal cannabis products from the black market"

Consider relying on neutral language, except in quotations, i.e. using the terms "illicit market" instead of "black market" throughout the text (e.g. P. 20).

This has been corrected throughout the article.

17. PP. 13 & 14, sentences almost similar: "The Medicinal Cannabis Users Association of Australia submission reported that where ‘cannabis clinics’ are available, often via telehealth consultation, they often charge "specialist" consultation rates and monitoring fees for which patients can rarely get a Medicare or Health Fund rebate (submission 9; page 7).", and "These clinics are charging fees to put in an application to the TGA that attracts no fee. They are charging "Specialist" consultation rates and monitoring fees for which patients can rarely get a Medicare or Health Fund rebate.” (Medical Cannabis Users Assoc of Australia, Submission 9, page 7)."

To remove ambiguity, the first sentence on Page 13 has been rephrased (pg. 13, text copied below). The sentence on Page 14 has not been changed as this is a quote from a submission.

“The Medicinal Cannabis Users Association of Australia submission reported that ‘specialist’ consultation and monitoring fees are regularly applied at ‘cannabis clinics’, often resulting in patients being rarely reimbursed through appropriate channels such as Medicare or other Health Fund rebates (Submission 9; page 7)”. 

18. P. 15: "At the time of this parliamentary inquiry (November 2019)" - I thought that it was between 17 January 2020 and 26 March 2020?

This has been rephrased (pg. 15, text copied below):

“At the time this parliamentary inquiry was called (November 2019)…”

19. P. 15: "compassionate access schemes"  "scheme"

This has been changed (pg. 15, text copied below):

“…have an established compassionate access scheme…”

20. P. 18: "However, Australia did not vote for additional recommendations made by the World Health Organisation, including the removal of border controls for Delta-9-tetrahydrocannabinol formulations" 

This is incorrect, or I did not properly understand the formulation.

A good article to understand the WHO recommendations is WHO’s first scientific review of medicinal Cannabis: from global struggle to patient implications https://www.emerald.com/insight/content/doi/10.1108/DHS-11-2021-0060/full/html

Australia voted in favour of all recommendations that were submitted to the vote, and was one of the few countries to do so. The recommendations that Australia did not vote in favour of were, in fact, not put on the ballot. See vote results here: “Report on the reconvened sixty-third session (2-4 December 2020), economic and social council official records, 2020 supplement no. 8A”, E/2020/28/Add.1, available at: undocs.org/E/2020/28/Add.1 and in https://ssrn.com/abstract=3932639. See also explanations of vote pp. 17-18 in: https://www.unodc.org/documents/commissions/CND/CND_Sessions/CND_63Reconvened/ECN72020_CRP24_V2007524.pdf where is stated:

"Australia supported recommendation 5.1 which recognizes that cannabis and cannabis resin are unlikely to produce ill-effects similar to other Schedule IV substances, and that some parties have enacted laws to enable the medical use of cannabis and cannabis resin.

− Australia considers this approach strikes an appropriate regulatory balance and simply recognizes that there is a legitimate medical use quite separate to the use of cannabis for research purposes.

Australia supported recommendation 5.2 and 5.3 Australia’s strong preference is for dronabinol, THC and its stereoisomers to be managed consistently with cannabis and cannabis resin under Schedule I of the 1961 Convention. This would remove ambiguity and complexity, especially as it applies to trade between member states.

Australia supported recommendation 5.4 which will improve consistency and remove ambiguity in the regulation of cannabis extracts and preparations.

Australia supported recommendation 5.5 on the basis that predominantly cannabidiol preparations with low concentrations delta-9 THC are of less risk of harm than those which contain greater than 0.2 per cent. This recommendation assists improving consistency and trade and reporting discrepancies."

Thank you to the reviewer for this clarification. We have reviewed the reference provided and update the manuscript to read as follows: 

p.g. 18 “In December 2020, Australia voted at the Commission on Narcotic Drugs for removal of cannabis and cannabis resin from Schedule IV of the Single Convention on Narcotic Drugs, 1961, and thereby recognising the use of cannabis for therapeutic purposes. Australia was one of only three countries to support all five recommendations submitted to vote by the World Health Organisation at the commission (ref). Border controls for Delta-9-tetrahydrocannabinol (dronabinol) formulations have been maintained, consistent with the current Schedule 8 poison status of cannabis that was implemented in Australia in 2017 [14].”

21. P. 18: "no product has met requirements to be supplied in this manner [16]". It is unclear if the reference mentioned related to the fact that no product has met requirements to be supplied in this manner. I would place the reference before the word "albeit", and slightly edit the sentence so it reads: "over the counter,[16] albeit no product has met requirements to be supplied in this manner *yet*."

This has been changed (pg. 18, text copied below):

“…down-scheduled to be available over the counter [16], albeit no product has met requirements to be supplied in this manner yet.”

22. P. 20: typo in "or by via self-cultivation".

This has been corrected (pg. 20, text copied below):

‘… or via self-cultivation.’

I look very much forward to reading an improved and final version of this article! Thank you for your work.

Thank you for this feedback.

 

Reviewer 2

A fascinating and well written article that qualitatively delves into over 120 submissions made by a variety of stakeholders to the 2019 Australian inquiry into barriers to patient access to medicinal cannabis. Using the modified Levesque patient-centred access to care framework, the authors examine these barriers from the perspective of appropriateness, availability and geographical accessibility, acceptability, and affordability. The analysis uncovers the tension and differences between government/professional and patient/advocacy stakeholders with regards to their perception of the current regulatory framework and the impact on access and use of medical cannabis. This manuscript provides a deep dive into the discourse surrounding access to medicinal cannabis in Australia and will be informative for other jurisdictions who have legalized/decriminalized access to medicinal cannabis.

Thank you for this feedback.

There are a few issues that would warrant further consideration and elucidation.

1. In the Introduction, a description of the SAS Category A and SAS-B pathways is presented – it is not clear if there is a distinction between “medical practitioners” and “registered health practitioners”. Given some jurisdictions allow other health professionals, such as nurse practitioners, to authorize/prescribe medical cannabis, it would be helpful to clarify if there is indeed a difference across these pathways re: type of health care professional.

The SAS allows prescribers (including nurse practitioners) to prescribe medicinal cannabis products for a single patient on a case-by-case basis. We have now clarified this in our revised manuscript. We have also replaced “medical practitioner” to “registered health practitioner” for consistency. 

2. There is reference throughout the manuscript to “scientific evidence” and what type and how much evidence is perceived to be required by different stakeholders to support the prescription of medical cannabis. The role of patient preference/values and contextual factors, such as the risks of other pharmaceutical treatments (e.g., opioids) and resources, however, is not mentioned. Given these are hallmarks of evidence-based medicine, it would be worthwhile for a brief discussion about whether evidence-based medicine is a guiding principle across government, professional and charity groups, as well as among the patient/advocacy stakeholders.

We thank the reviewer for the comment. We have now revised the manuscript according to reviewer’s comment. see the text below. 

“While the need to incorporate patients' preferences into medical decision making has been highlighted by various stakeholders as a key component of patient-centred care, this has not been translated to decisions around access to and use of medical cannabis. For example, some patients, families, and caregivers reported using medical cannabis due to the significant financial distress, poor symptom control or intolerable adverse effects from conventional therapies (e.g., opioid dependence). One patient described medical cannabis as “a gateway out of the hopelessness of opioid dependence… alcoholism and addiction to prescription benzos”. Yet, what patients described as their lived experiences was considered by many stakeholders as ‘anecdotal’ evidence, and it was argued that they should not be taken into consideration in the policy making process.”

3. I was intrigued by the quote drawn from the health and medical field that suggested that “We firmly believe that patients deserve access to medicines of the highest quality – testing for both safety and efficacy in placebo-controlled, randomized clinical trials and licensed as medicines…”. This quote appears quite naïve of the reality of medical cannabis research with regards to limited funding as well as the challenges associated with conducting clinical trials on a plant-based medicine that is difficult standardize as well as blind subjects to due to its psychoactive effect. It also expresses a clear bias toward the medicalization of cannabis, assuming that medicinal cannabis should be developed into a pharmaceutical medicine that can be patented and licensed. I would encourage the authors to reflect further in the discussion on the tension that exists when these two different paradigms of therapies (plant vs. pharmaceutical) collide.

The specific quote “We firmly believe that patients deserve access to medicines of the highest quality – testing for both safety and efficacy in placebo-controlled, randomized clinical trials and licensed as medicines…” came from cannabis/pharmaceutical industry, and do not necessarily reflect the views of “health and medical field”. We have therefore replaced the quote from GQ Pharmaceuticals to a representative quote from a health and medical profession (as suggested by reviewer 1). The need for clear and convincing evidence from gold standard RCTs, and the difficulty of achieving such data is highlighted in the revised manuscript. 

4. The discussion would have benefitted from a brief discussion of other medical cannabis regulatory systems, including in Canada and the US, and whether the associated policies/processes would address some of the barriers expressed by stakeholders in Australia.

Globally, the laws and policy domains in the majority of countries are grounded in and shaped by the existing and historical cannabis regulation making it difficult to draw credible comparisons across countries. However, we have tried to describe this, and identify areas commonalities across countries. 

“Internationally, the question of how to regulate medical cannabis is a major source of debate in many countries and continues to divide the public health community. The laws and policy domains in the majority of countries are grounded in and shaped by the existing and historical cannabis regulation [30], making it difficult to draw credible comparisons across countries. Despite differences in their regulatory approaches to medical cannabis, majority of these countries agree on: i) the need for continued education for health practitioners, ii) addressing stigma, iii) involving patients and addressing patients’ concerns, and iv) improving the evidence base through real-world data collection.”

5. Recommend that not only do medical and pharmacy schools need to include education on the ECS and medicinal cannabis, but also nursing schools as nurses are often tasked in clinical settings to assist patients in using medicinal cannabis. To do so, they require sufficient education about medicinal cannabis.

This has been included (pg. 17, text copied below):

“…revising of curriculum of medical, nursing and pharmacy schools to include endocannabinoid system…”

6. In the Discussion, there is commentary regarding the acceptability of medical cannabis being viewed as an attempt to ensure patient safety; however, in restricting access and pushing patients towards the unregulated/illegal market, they are also placing them at risk through the use of an unregulated supply of cannabis as well as the risk of criminalization. Some discussion of this is required.

7. The need to “decouple” medicinal cannabis from that used for recreational purposes is indeed important; however, it would be useful to suggest some strategies to do (i.e., education, public education, working with professional societies with stigmatizing policies).

We have now addressed these comments as below. 

“Given that these stigmas toward therapeutic use of cannabis are mainly due to its connections to recreational cannabis, it may partially be addressed by decoupling recreational vs medical cannabis via increased education and public awareness campaigns, as recommended by the Senate Inquiry committee. While concerns regarding the acceptability of medical cannabis may equally be viewed as attempting to ensure patient safety, they conversely result in making access to medical cannabis more difficult for consumers. Evidence strongly suggests that the complex regulatory framework for accessing medical cannabis is contributing to some people choosing to access illicit cannabis products, which has its own quality and safety implications and legal repercussions.”

Minor grammatical/writing issues:

8. The term “peak health bodies” is unique to Australia and does not easily translate for readers from other countries. Please define when first used and provide an example.

“Peak health body” is the term commonly used in Australia to refer to “professional health bodies” such as Australian Medical Association. In order to avoid confusion, we have replaced “peak” with “professional” throughout the manuscript. 

9. The word “neogitations” is spelled wrong on the 3rd line above the section entitled “The role of evidence in informing policy and clinical practice…”

This has been corrected. 

10. Remove “an” from the following sentence, “…and advocated for an expedited access while TGA registration is pending.”

This has been corrected.

11. In the first sentence under “Supply – demand mismatch”, edit to read: “…Australian made products, along with the fact that imported medicinal cannabis…”

This has been corrected.

12. Edit the following sentence in the discussion: “…cost that patients would otherwise pay at cannabis access clinics…”

This has been corrected.

13. There are too many acronyms, particularly in the Discussion. Please consider keeping only a handful of them so the reader is not confused or searching for what they stand for throughout the manuscript.

We have now reduced the number of acronyms throughout the manuscript. 

14. The sentence with “This ‘postcode’ lottery is evidence “appears to be unfinished in the Discussion.

We have now deleted this incomplete sentence since it was a repetition. 

15. The term “black market” has fallen out of favour; please consider using “unregulated” or “illegal” market instead.

This has been corrected throughout the article. Please refer to Comments 13 and 16 above from Reviewer 1.

---

## [Decision Letter · Decision Letter 1]

26 Oct 2022

From growers to patients: multi-stakeholder views on the use of, and access to medicinal cannabis in Australia

PONE-D-22-15976R1

Dear Dr. Erku,

We’re pleased to inform you that your manuscript has been judged scientifically suitable for publication and will be formally accepted for publication once it meets all outstanding technical requirements.

Kind regards,

David James Carter, PhD, LLM (Res), LLB (Hon I), BA

Academic Editor

PLOS ONE

Additional Editor Comments (optional):

Thank you for your careful and considered response to reviewer feedback. I have reviewed both Reviewer 1 and Reviewer 2's comments on your first revision and am happy that you have been able to respond to their feedback in a timely and careful manner.

I note that Reviewer 2 has provided a final comment regarding a typographical/consistency issue in their revision feedback. I believe that this can be addressed during the production and proof revision process and is not the basis for a minor revision decision at this point in time. However, those who manage the production workflow may set this to be a minor revision to facilitate that change should that be more efficient in terms of workflow.

The feedback which must be addressed in proofing stages is as follows: First paragraph on p. 6 – there is still reference to physicians that imply they are the sole prescribers of medical cannabis in Australia – please include NPs or make more generic. This again occurs on p. 11, 13, 15. Please ensure both physicians and NPs are acknowledged. Academic Editor comment: This should be managed during production processes. In this instance it would be suitable to simply refer to authorised prescribers/prescribers/registered health practitioners who are able to prescribe etc rather than simply a reference to NP and medical practitioners.

Reviewers' comments:

Reviewer's Responses to Questions

**Comments to the Author**

1. If the authors have adequately addressed your comments raised in a previous round of review and you feel that this manuscript is now acceptable for publication, you may indicate that here to bypass the “Comments to the Author” section, enter your conflict of interest statement in the “Confidential to Editor” section, and submit your "Accept" recommendation.

Reviewer #1: All comments have been addressed

Reviewer #2: (No Response)

2. Is the manuscript technically sound, and do the data support the conclusions?

Reviewer #1: Yes

Reviewer #2: Yes

3. Has the statistical analysis been performed appropriately and rigorously? 

Reviewer #1: N/A

Reviewer #2: Yes

4. Have the authors made all data underlying the findings in their manuscript fully available?

Reviewer #1: Yes

Reviewer #2: Yes

5. Is the manuscript presented in an intelligible fashion and written in standard English?

Reviewer #1: Yes

Reviewer #2: Yes

6. Review Comments to the Author

Reviewer #1: Many thanks, I appreciated the comments of the other reviewer and your smooth update of the manuscript taking comments from the two of us into account. Best wishes.

Reviewer #2: First paragraph on p. 6 – there is still reference to physicians that imply they are the sole prescribers of medical cannabis in Australia – please include NPs or make more generic. This again occurs on p. 11, 13, 15. Please ensure both physicians and NPs are acknowledged.

Otherwise, thank you for the careful consideration of the reviewers’ comments, the authors did an excellent job of addressing.

7. PLOS authors have the option to publish the peer review history of their article (what does this mean?). If published, this will include your full peer review and any attached files.

Reviewer #1: No

Reviewer #2: No

---

## [Editor Report · Acceptance letter]

3 Nov 2022

PONE-D-22-15976R1 

From growers to patients: multi-stakeholder views on the use of, and access to medicinal cannabis in Australia 

Dear Dr. Erku:

I'm pleased to inform you that your manuscript has been deemed suitable for publication in PLOS ONE. Congratulations! Your manuscript is now with our production department. 

Kind regards, 

on behalf of

Dr. David James Carter 

Academic Editor

PLOS ONE